# The Prevalence of Overweight, Obesity, Hypertension, and Diabetes in India: Analysis of the 2015–2016 National Family Health Survey

**DOI:** 10.3390/ijerph16203987

**Published:** 2019-10-18

**Authors:** Vishal Vennu, Tariq A. Abdulrahman, Saad M. Bindawas

**Affiliations:** Department of Rehabilitation Sciences, College of Applied Medical Sciences, King Saud University, 10219 Riyadh 11433, Saudi Arabia; tabdulrahman@ksu.edu.sa (T.A.A.); sbindawas@ksu.edu.sa (S.M.B.)

**Keywords:** obesity, hypertension, diabetes, prevalence, rate, urban, rural, India

## Abstract

Overweight, obesity, hypertension, and diabetes increase the risk of non-communicable diseases and all-cause mortality worldwide. Previous studies have not determined the prevalence of these conditions/diseases throughout India. Therefore, this study was aimed to address this limitation. Data on these conditions/diseases among men and women aged ≥ 18 years were obtained from the fourth National Family Health Survey conducted throughout India between January 2015 and December 2016. The prevalence and prevalence rate per 100,000 population were calculated at the national level and by age group, sex, and type of residence for each state and union territory. The national prevalence of overweight, obesity, hypertension, and diabetes were 14.6%, 3.4%, 5.2%, and 7.1%, respectively. The highest prevalence of these conditions/diseases at the national level was seen among those aged 35–49 years (54 years for men), especially women living in urban areas. In India, 1 out of every 7, 29, 19, and 14 individuals at the national level had overweight, obesity, hypertension, and diabetes, respectively—between 2015 and 2016. These results are important for the healthcare system and government policies in the future. Moreover, targeted efforts are required to establish public health strategies for the prevention, management, and treatment of these conditions/diseases throughout India.

## 1. Introduction

India is the second most populous developing country with industrialization, and rapid urbanization has resulted in a significant number of people being overweight or obese and/or having elevated blood pressure and blood glucose [1,2]. These conditions/diseases increase the risk of non-communicable diseases [3]. Moreover, the coexistence of these conditions/diseases causes a considerable rise in disease risk [4]. It is well established that poor health outcomes and all-cause mortality, approximately 2.8 million cases (1 in 6 individuals) annually, are attributed to overweight, obesity, hypertension, and diabetes [5], possibly because these conditions/diseases increase the likelihood of coronary heart disease, stroke, certain cancers, obstructive apnea, and osteoarthritis [5].

Overweight and obesity are considered to be significant and increasing public health problems worldwide [6]. The prevalence of overweight and obesity (ranging from 26% to 3%, respectively) and associated deaths (6.5%) in adults are rising rapidly in several countries [3,7], including India [6,8]. The 2016 World Health Organization (WHO) report described overweight/obesity as a pandemic in India because of a 3.4% increase in prevalence from 2006 (8.4%) to 2016 (11.8%) [6]. A previous review also reported an overall increase in the prevalence of overweight and obesity between 1998 and 2006, from 11% to 15% in men and women aged 15–49 years [2].

One of the major treatable public health problems on the rise is hypertension, and it is linked to both overweight and obesity [9]. Hypertension underlies the development of cardiovascular diseases, other non-communicable diseases, and mortality (32.5%), both globally and in India [9,10,11,12]. A recent review revealed that the overall prevalence of hypertension in India was 29.8%, and it varies between rural (27.6%) and urban (33.8%) populations [13]. Another community-based study demonstrated that the prevalence of hypertension was higher in urban than in rural areas [14].

The most common chronic disease worldwide is diabetes mellitus, and its prevalence has risen from 4.7% (108 million) in 1980 to 8.5% (422 million) in 2014 among adults aged ≥ 18 years [15]. Approximately, 439 million adults are estimated to have diabetes by 2030 [16]. The highest increase in prevalence is expected to occur in low- and middle-income countries, including India [15,17]. In 2010, India had the highest number of people with diabetes, around 50.8 million; this number is set to increase to 87 million by 2030 [18]. Every year, about 1 million deaths in India are attributed to diabetes; it is also the primary reason for blindness, kidney failure, heart attack, stroke, and lower limb amputation [19].

Numerous earlier studies have estimated the prevalence of overweight, obesity, hypertension, and diabetes among adults in rural and urban India [20,21,22,23,24]. However, these studies were focused on certain parts of India and used various methodologies and populations. Moreover, different cut-off points for determining overweight, obesity, hypertension, and diabetes were used in these studies. A recent study examined trends in the prevalence of overweight/obesity by socioeconomic position between 1998 and 2016 using data from the national family health surveys conducted in India. However, that study was limited to 1 condition (overweight/obesity) in a different population aged ≥ 15 years with a body mass index (BMI) cut-off score of 24.99 kg/m^2^. Furthermore, prevalence data by rural and urban areas for each of the Indian states are not presented [8]. Thus, the authors acknowledged the absence of studies that estimated the prevalence of overweight, obesity, hypertension, and diabetes in all 36 Indian entities, comprising 29 states and 7 union territories (UTs), according to age group, sex, and type of residence [8,17,21,22,24].

The purpose of this study was to address the above-mentioned shortcomings by determining the prevalence of overweight, obesity, hypertension, and diabetes (i) at the national level by age group, sex, and type of residence; (ii) per age group for each Indian state and UT; (iii) per age group, sex, and type of residence for each Indian state and UT; and (iv) at the national level (overall prevalence rates) and for each Indian state and UT.

## 2. Materials and Methods 

The flow of the current study sample is illustrated in Figure 1. Of the total 699,686 eligible respondents of the fourth National Family Health Survey (NFHS-4), 699,481 adults aged ≥ 18 years were included in the current study after excluding individuals aged younger than 18 years (*n* = 205). According to the current study sample ages, we classified respondents into two age groups: 18–34 years (*n* = 104,559) and 35–49 years (54 years for men) (*n* = 594,922).

The datasets generated and analyzed during the current study are available in the National Family Health Survey repository [25]. The NFHS-4 provides data on population health and nutrition for each state and UT in India. The primary objective of the NFHS-4 is to provide essential data on health, family welfare, and related issues in these areas. The NFHS-4 was designed to offer vital estimates of the prevalence of clinical, anthropometric, and biochemical components, such as malnutrition, anemia, hypertension, HIV, and high blood glucose levels, through a series of biomarker tests and measurements.

In addition to the 29 states, the NFHS-4 also included all 7 UTs for the first time, including 640 districts covered by the 2011 census. However, this survey was conducted under the stewardship of the Ministry of Health and Family Welfare and coordinated by the International Institute for Population Sciences (IIPS), Mumbai. The IIPS co-operated with many field organizations to conduct the survey in all the Indian states and UTs. Technical assistance for the NFHS was provided mainly by ICF International Inc., Virginia, the United States and other organizations on specific issues, such as problems with mini-laptops used for data entry and data transfer to the IIPS. Details about the NFHS-4 sample design, questionnaires, biomarker measurements, and tests (Appendix A) are available elsewhere [26].

In the present study, data on a particular biomarker, such as body mass index (BMI), blood pressure, and blood glucose among adult men (aged 18–54 years) and women (aged 18–49 years), were included from any 1 of the many biomarkers of the NFHS-4. The questions and techniques used to assess the presence of overweight, obesity, hypertension, and diabetes among adults are shown in Box 1. The Institutional Review Board of the IIPS and ICF approved the protocol for the NFHS-4, including the content of all the survey questionnaires. Informed consent was obtained from all the participants who accepted participation in the survey.

Box 1Questions and procedures followed for identifying the presence of overweight, obesity, hypertension, and diabetes among adults under the National Family Health Survey 2015–2016.**Body Mass Index, kg/m^2^:** Respondent’s height in centimeters (cm) and weight in kilograms (kg) were measured using the SECA 213 Stadiometer (SECA Inc., Hamburg, Germany) and SECA 874 U digital scale (SECA Inc., Birmingham, UK), respectively. BMI was estimated by dividing the weight in kilograms by the height in meters squared (kg/m^2^). Overweight and obesity were said to exist (and the data included in this study) if the BMI was 25 to <30 or ≥30 kg/m^2^, respectively, based on the World Health Organization definition [27]. Underweight (BMI < 18 kg/m^2^) and normal weight (BMI = 18 to <25 kg/m^2^) were excluded from the analysis.**Blood pressure (BP), mm Hg:** Will you allow me to measure your blood pressure? You can say yes or no to the test. You are free to decide. Circle the appropriate code and sign your name: 1 = Granted, 2 = Refused, 3 = Granted (no signature). For those who circled 1 or 3, the following questions were asked to identify factors that may affect BP measurements: Have you done any of the following things in the past 30 minutes? Have you had anything to eat? Eaten: 1 = Yes, 2 = No. Have you had coffee, tea, cola, or another drink that contains caffeine? Had a caffeinated drink: 1 = Yes, 2 = No. Have you smoked tobacco or any tobacco product? Smoked: 1 = Yes, 2 = No. Have you used another type of tobacco such as ghutka, pan masala with tobacco, other chewing tobacco, or snuff? Other tobacco: 1 = Yes, 2 = No. The following question was asked to those who answered ‘no’ to the previous questions: May I measure your blood pressure? If they said ‘yes,’ the circumference of the arm between the elbow and the shoulder was measured in centimeters (cm) to ensure that the right equipment was used. The arm circumference measurement was used to select the appropriate BP monitor cuff size by circling 1 = small (17–22 cm), 2 = medium (22–32 cm), or 3 = large (32–42 cm). The BP (systolic and diastolic) reading (3 measures in total) were recorded with the help of the OMRON Blood Pressure Monitor (OMRON Healthcare, Hoofddorp, Netherlands). If BP was not measured, other survey questions were asked. If BP was measured 3 times with a 5-min break between readings, a single number where the average diastolic and systolic measures met was circled:**Average Diastolic****Average Systolic****<80****<85****85–89****90–99****100–109****≥110**<120123456<130123456130–139123456140–159123456160–179123456≥180123456
Where 1 = normal (optimal), 2 = normal (mildly high), 3 = normal (moderately high), 4 = abnormal (mildly elevated), 5 = abnormal (moderately elevated), and 6 = abnormal (severely elevated). If only 2 measurements were taken, the second systolic and diastolic numbers were recorded. If only 1 measurement was made, the first systolic and diastolic numbers were recorded. Abnormal BP readings (moderately to severely elevated) were included in the present study.**Blood glucose, mg/dL:** Do you have any questions about the blood sugar measurement so far? If you have any questions about the procedure at any time, please ask me. You can say yes or no to having your blood sugar measured now. Will you allow me to take your measurement? Circle the appropriate code and sign your name: 1 = Granted, 2 = Refused, and 3 = Granted (no signature). If 1 or 3 was circled, the following questions were asked: When was the last time you had something to eat? When was the last time you had something to drink other than plain water? If they answered less than 1 h, “00” was recorded. If they answered hours ago, the blood glucose level in mg/dL was recorded after collecting the finger-stick blood specimen and conducting a test using the Optimum H Glucometer (Abbott Laboratories, Chicago, IL, USA). The readings were considered equivalent to laboratory estimations of blood glucose levels made using the glucose oxidase method for glucose levels in the range of 10–600 mg/dL. Diabetes was said to exist (and the data included in this study) in adults who had blood glucose levels ≥ 140 mg/dL. Those who refused the test and those with blood glucose levels under 140 mg/dL were excluded.

Descriptive statistics of means and standard deviation were computed for continuous variables, whereas frequencies (percentage) were calculated for categorical parameters. The national prevalence of overweight, obesity, hypertension, and diabetes was computed by age group, sex, and type of residence as well as for all Indian states and UTs. Further, the prevalence rates of overweight, obesity, hypertension, and diabetes per 100,000 population were calculated at the national level and for all Indian states and UTs. All prevalence results were computed using Statistical Analysis Software (SAS) version 9.2 (SAS Institute Inc., Cary, NC, USA), operated on Windows.

## 3. Results

The sociodemographic characteristics of the current study sample are shown in Table 1. The majority of the individuals were in the 35–49-years age group (54 years for men) with a mean age of 47.6 ± 12.8 years. Most of the individuals were women (86.5%), unemployed (76.6%), residing in a rural area (70.7%), and did not have any insurance (82.1%). Moreover, the majority of the respondents were Hindus (74.1%).

The national prevalence of overweight, obesity, hypertension, and diabetes according to age group, sex, and type of residence is shown in Figure 2. Between 2015 and 2016, overweight (14.6%) was more prevalent nationally than obesity (3.4%), followed by diabetes (7.1%) and hypertension (5.2%) (Figure 2A). Compared to individuals aged 18–34 years, men, and rural areas, a high prevalence of these conditions/diseases was seen among individuals aged 35–49 years (54 years for men) (Figure 2B), women (Figure 2C), and urban areas (Figure 2D), respectively.

Overweight, obesity, hypertension, and diabetes prevalence per 100,000 population at the national level and per state and UT are shown in Figure 3, Figure 4, Figure 5 and Figure 6, respectively. Compared to other states and UTs, Uttarakhand had a high prevalence per 100,000 population of overweight (14,620) which was above the national average (14,610) (Figure 3). West Bengal (6400) followed by Uttarakhand (5095), Uttara Pradesh (4190), Tripura (3825), Telangana (3750), and Tamil Nadu (3670) had a high prevalence of obesity which was also above the national average (3420) (Figure 4). The Andaman and Nicobar Islands had the lowest prevalence of both overweight (3915) and obesity (655). The prevalence of hypertension in each state and UT was below the national average (5170). Assam had the highest prevalence of hypertension (5155), whereas Delhi had the lowest rate (1480) (Figure 5). While Delhi had the highest prevalence (10,390) of diabetes above the national average (7060), the lowest prevalence was seen in Rajasthan (2210) (Figure 6).

The prevalence of overweight, obesity, hypertension, and diabetes among adults according to age group, sex, and type of residence for each state and UT are summarized in Figure 2, Figure 3, Figure 4 and Figure 5, respectively. Compared to 36 states and UTs, men aged 18–34 years living in urban areas of the Andaman and Nicobar Islands (28.6%) had the highest prevalence of overweight (Table 2). Women aged 35–49 years living in rural areas of Lakshadweep (12%) had the highest prevalence of obesity (Table 3). Men aged 35–54 years residing in the Arunachal Pradesh urban region had the highest prevalence (11.3%) of hypertension (Table 4). Further, men aged 35–54 years living in the urban area of Delhi (21.3%) had the highest prevalence of diabetes (Table 5).

## 4. Discussion

The present study was designed to determine the prevalence of overweight, obesity, hypertension, and diabetes at the national level and for 29 states and 7 UTs according to age group, sex, and type of residence. The study revealed that the disease/condition with the highest prevalence in India between 2015 and 2016 was overweight followed by diabetes, hypertension, and obesity. At the national level, the 35–49-years (54 years for men) age group, women, and urban areas had a greater prevalence of these conditions/diseases than individuals aged 18–34 years, men, and rural areas. At the state and UT level, Lakshadweep residents aged 35–49 years (54 years for men) had a higher prevalence of overweight and obesity. Nagaland and Delhi residents aged 35–49 years (54 years for men) had a high prevalence of hypertension and diabetes, respectively. The study also revealed that the prevalence of overweight was high and above the national average among residents of Uttarakhand. Moreover, rates of obesity above the national average were seen in people from West Bengal, Uttarakhand, Uttara Pradesh, Tripura, Telangana, and Tamil Nadu. The lowest rates of overweight and obesity were found in people from the Andaman and Nicobar Islands. 

The prevalence of hypertension in each state and UT was below the national average. However, the highest prevalence of hypertension was seen in residents of Assam, whereas the lowest prevalence was found in residents of Delhi. While the highest rate of diabetes was seen in Delhi, the lowest rate was observed in Rajasthan. Surprisingly, the findings showed that the highest prevalence of overweight was among men aged 18–34 years living in the urban areas of the Andaman and Nicobar Islands. Another important finding was that the highest prevalence of obesity, hypertension, and diabetes was seen in the 35–49-years (54 years for men) age group and in women living in urban areas of Lakshadweep and Arunachal Pradesh and in men from Delhi, respectively.

The results of this study are compatible with the WHO estimates [5] and a recent NFHS-4 report [28] on the prevalence of overweight and obesity at the national level and in men and women living in rural and urban India [8,17,21,22,24]. In accordance with the present results, the NFHS-4 report demonstrated that overweight and obesity occur at a rate of 15.5% and 5.1%, respectively, at the national level in India. However, the results have not previously been described according to age groups, including for each state and UT [8,29,30,31], and contradict previous findings [32]. It seems possible that these results of the current study differ due to the classification of the age groups. 

To our knowledge, this is the first cross-sectional study that assessed the prevalence of overweight, obesity, hypertension, and diabetes at the national level and for each state and UT according to age group, sex, and type of residence. In this study, the results showed that the prevalence of overweight and obesity was the highest in Lakshadweep residents aged 35–49 years (54 years for men). This finding is consistent with that of Ahirwar and Mondal’s systematic review (2019) in which they found that the prevalence of obesity was high in men (24.6%) and women (41.4%) from Lakshadweep. A possible explanation for this finding is that a sedentary lifestyle and unhealthy dietary pattern have been reported previously for Lakshadweep [33]. It is somewhat surprising that a high prevalence of overweight was observed among men aged 18–34 years living in urban areas of the Andaman and Nicobar Islands. This finding is corroborated by a previous study that provided evidence of a progressive increase in overweight among adults in the Andaman and Nicobar Islands [34]. According to the results of that study, the prevalence of overweight has increased to 22% among men and women since 1960. Another important finding was that the prevalence of overweight in Uttarakhand was high and above the national average. A possible explanation for this result may be changed in lifestyle [35].

A notable finding of the present study was that the prevalence of hypertension was high in Nagaland residents aged 35–49 years (54 years for men) and urban areas of Arunachal Pradesh. The high prevalence of hypertension in Nagaland residents aged ≥ 35 years may be attributed to socioeconomic development, changes in lifestyle and dietary intake, and food consumption patterns [36]. The high rate of hypertension in urban areas of Arunachal Pradesh may be attributed to the high percentage of smoking and tobacco chewing among the population [37]. Moreover, the higher rate of hypertension in people from Assam may be attributed to the increased consumption of salt, the intake of locally prepared alcohol, increased BMI, and central obesity [38]. A community-based study on older individuals revealed BMI, higher education status, and diabetes to be significantly associated with the prevalence of hypertension [14]. However, the determined rate of hypertension in the current study was lower than that in other studies [13,14,20]. Recent systematic reviews estimated a rising burden of hypertension in India [13,39]. The findings from the systematic review and meta-analysis by Anchala and colleagues indicated that the overall prevalence of hypertension in India was 29.9% after weighting the regional population size [13]. Further, the findings of that review revealed that the prevalence of hypertension was high in urban areas and differed significantly compared to rural areas. Results from the analysis of a nationally-representative survey showed that women had high rates of hypertension compared to men and it increased steeply with BMI [20].

The results of this study showed for the first time that the majority of the men with diabetes were from the urban region of Delhi. It is difficult to explain the high prevalence of diabetes in Delhi; however, it might be related to changes in dietary habits, energy expenditure, competition, and improved living conditions [40]. Previous studies have suggested that there is a relationship between diabetes and high white rice consumption [41], daily or weekly fish intake [42], BMI, age, waist-to-hip ratio, family history of diabetes, monthly income, and sedentary physical activity [41,42,43,44].

The current results related to diabetes were consistent with previous studies that reported a higher prevalence of diabetes among adults in urban (11.2%) compared with rural (5.2%) India [17,22,45]. The third NFHS (NFHS-3), which covered more than 99% of the population except for the small UTs, reported that the prevalence of diabetes was higher in men (1598 per 100,000 people) than in women (1054 per 100,000 people) [42]. The NFHS-3 also reported a variation in rural-urban and geographic regions with higher rates in south and north-eastern India. An earlier National Urban Diabetes Survey covering all the regions of India reported that the national age-standardized prevalence of diabetes was 12.1% and that it was high in urban India [43].

In countries like India, the prevalence rates of overweight, obesity, hypertension, and diabetes could be attributed to the remarkable shifts in the nutritional scenario over the past 7 decades [46,47]. Further, industrialization and rapid urbanization in all states and UTs of India increased the indulgence in habits such as smoking, chewing tobacco, drinking alcohol, using mechanized transport and technology, and watching television alongside a tendency toward sedentary lifestyles with high physical inactivity resulting in an increase in the occurrence of overweight and obesity [46]. This scenario is associated with many conditions, including hypertension and diabetes [3,4], which pose a severe problem for the health and quality of life of adults in India. Therefore, cooperation at a multidisciplinary level, such as involving all the stakeholders including the central and state governments, non-government organizations, researchers, and healthcare professionals at large, is necessary for the implementation of policies and programs to improve the well-being and quality of life of residents of India.

To its credit, this study has several strengths. The main strength of this study is that the data were obtained from the NFHS-4, the first large-scale, multi-round survey that conducted and collected data with a high response rate in rural and urban areas across all 36 entities of India using standard protocols and quality control procedures. Second, all the field teams of the NFHS-4 were well constructed with qualified and trained professionals supported by state-of-the-art technology. Third, overweight, obesity, hypertension, and diabetes were well-defined and measured and tested with equipment for quality and accuracy. Finally, the prevalence results of overweight, obesity, hypertension, and diabetes are presented for the first time in this study for each state and UT of India according to age group. However, the study also had a few limitations. First, the cross-sectional nature of the design did not allow for the cause-effect relationship to be determined effectively. Second, since this was secondary data analysis, there is a likelihood of selection bias. Third, this study did not include data to examine the predictors or risk factors of overweight, obesity, hypertension, and diabetes. Finally, overweight and obesity were computed from BMI, rather than the alternative measures of waist circumference and body fat percentage [48].

## 5. Conclusions

This study was set out to determine the prevalence of overweight, obesity, hypertension, and diabetes at the national level and for each state and UT in India according to age group, sex, and type of residence. The results of this study showed that India was an overweight state of the nation, followed by diabetes, hypertension, and obesity, respectively, between 2015 and 2016. Individuals aged 35–49 years (54 years for men) living in urban areas, especially in Lakshadweep, Nagaland, and Delhi, had the highest prevalence of these conditions/diseases compared to those aged 18–34 years living in rural areas. While Uttarakhand had the highest rate of overweight with more than the national average, West Bengal, Uttarakhand, Uttara Pradesh, Tripura, Telangana, and Tamil Nadu had higher rates of obesity than the other states or UTs of India. The highest prevalence of hypertension was seen in the urban areas of Arunachal Pradesh and Assam. Our findings should be a matter of great concern warranting urgent preventive measures at a multidisciplinary level, involving the central and state governments, non-governmental organizations, and healthcare professionals, to implement policies and programs to improve the well-being and quality of life of individuals at the regional and national levels. Further studies should be conducted using waist circumference and body fat percentage to assess overweight and obesity and examine the risk factors for overweight, obesity, hypertension, and diabetes throughout India. 

## Figures and Tables

**Figure 1 ijerph-16-03987-f001:**
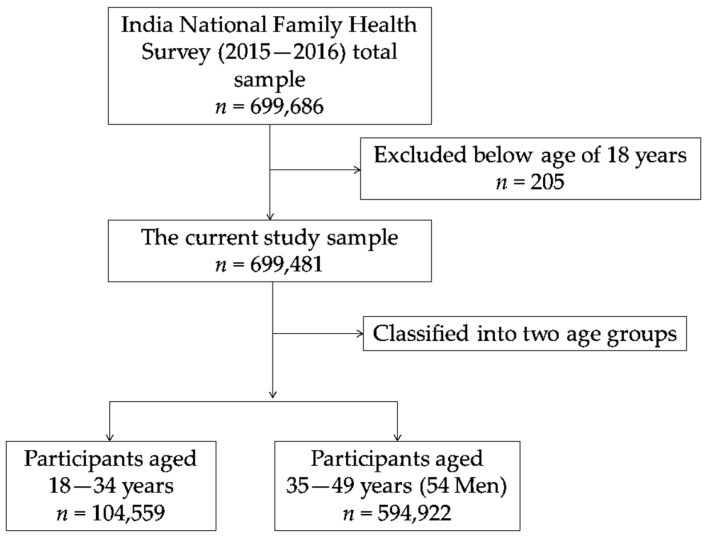
The flowchart of the current study sample.

**Figure 2 ijerph-16-03987-f002:**
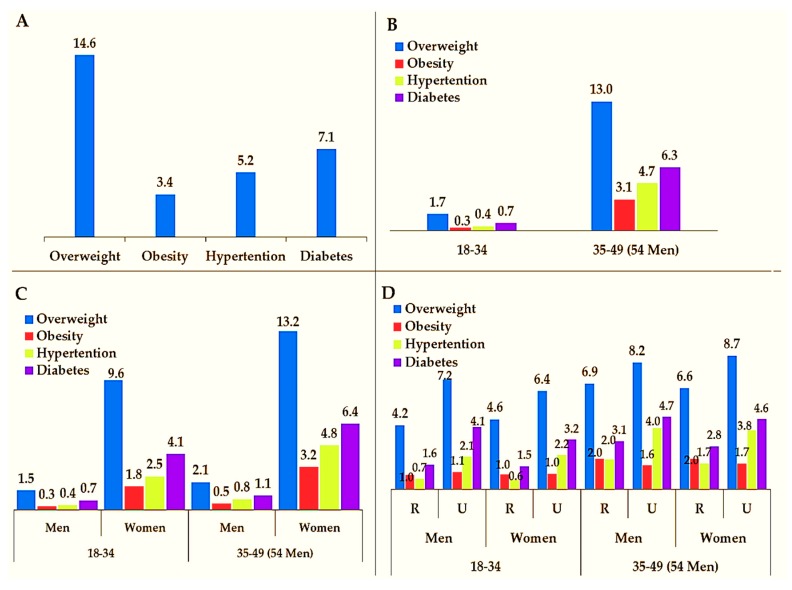
The national prevalence of overweight, obesity, hypertension, and diabetes among adults by (**A**) disease/condition, (**B**) age group, (**C**) age group and sex, and (**D**) age group, sex, and residence. The prevalence presented in percentages.

**Figure 3 ijerph-16-03987-f003:**
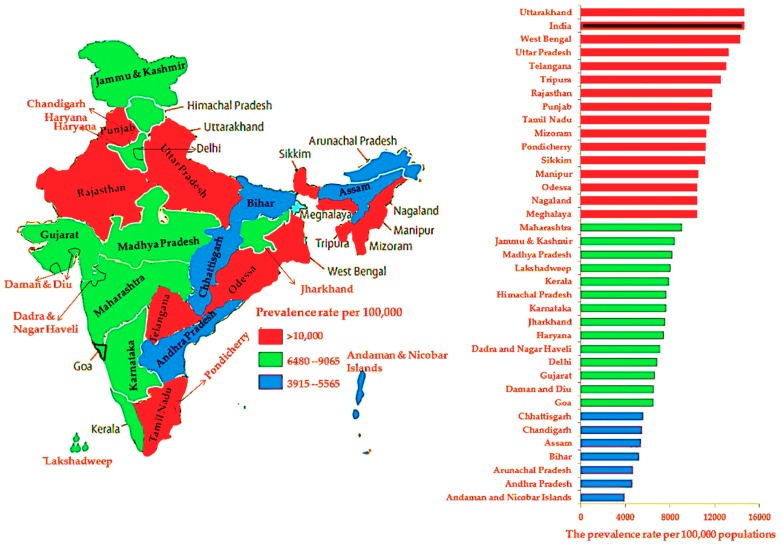
The prevalence rates of overweight among adults at the national level and per state and union territory.

**Figure 4 ijerph-16-03987-f004:**
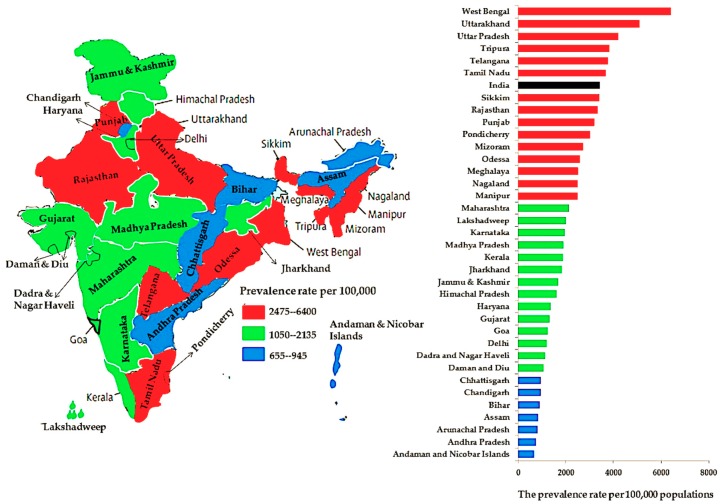
The prevalence rates of obesity among adults at the national level and per state and union territory.

**Figure 5 ijerph-16-03987-f005:**
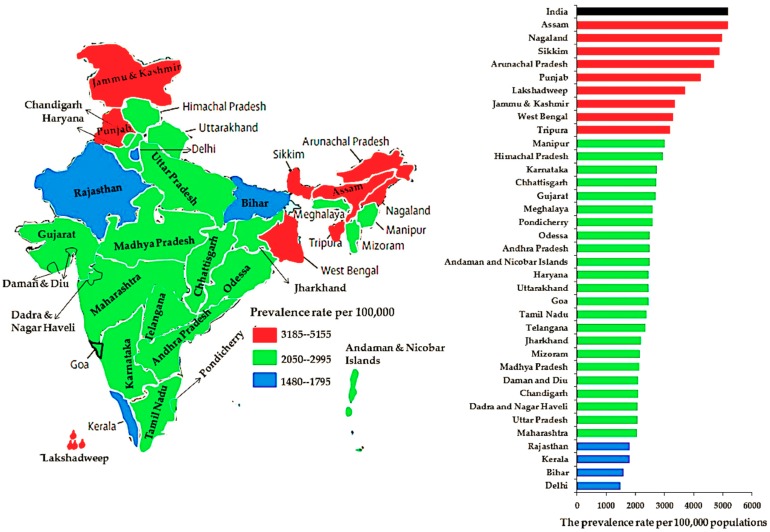
The prevalence rates of hypertension among adults at the national level and per state and union territory.

**Figure 6 ijerph-16-03987-f006:**
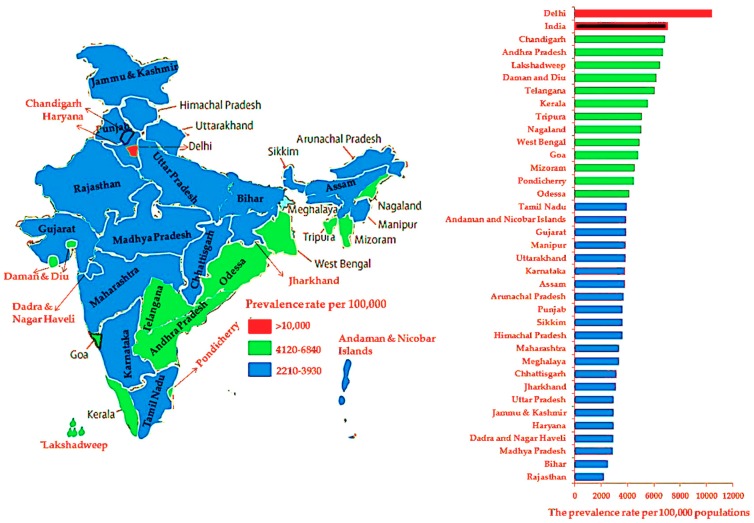
The prevalence rates of diabetes among adults at the national level and per state and union territory.

**Table 1 ijerph-16-03987-t001:** General characteristics of the study sample.

Characteristics	Mean ± SD OR *n* (%)
Age in years	47.6 ± 12.8
**Age group**	
18–34	104,559 (15)
35–49 (54 Males)	594,922 (85)
**Sex**	
Men	94,774 (13.5)
Women	604,912 (86.5)
**Place of residence**	
Rural	494,951 (70.7)
Urban	204,735 (29.3)
**Educational status**	
No education	1012 (0.5)
Primary	30,299 (15.6)
Secondary	51,852 (26.6)
Higher	111,640 (57.3)
**Marital status**	
Never married	171,797 (24.5)
Married	499,627 (71.4)
Widowed/diverse/separated	28,262 (4.1)
**Working status**	
Unemployed	93,713 (76.6)
Employed	28,638 (23.4)
**Religion**	
Hindu	519,281 (74.2)
Muslims	94,591 (13.5)
Christians	52,113 (7.5)
Sikhs	15,300 (2.2)
Buddhists/Neo-Buddhists	8981 (1.3)
Jainism	1028 (0.1)
Others/no-religion	8378 (1.2)
**Insurance status**	
No	574,718 (82.1)
Yes	124,968 (17.9)
**Clinical Indicators**	
Body mass index (BMI), kg/m^2^	26.2 ± 5.1
Blood pressure, mmHg	222.9 ± 15.7
Glucose level, mg/dL	156.8 ± 16.3

**Table 2 ijerph-16-03987-t002:** The prevalence of overweight among adults according to age group, sex, and type of residence for each state and union territory.

State/Union Territory (Alphabetical Order)		18–34 Years						35–49 (54 Men) Years					
		Men			Women			Men			Women		
		Total	Rural	Urban	Total	Rural	Urban	Total	Rural	Urban	Total	Rural	Urban
**1**	Andaman & Nicobar Islands	15.3	2.0	28.6	11.5	4.3	18.8	11.1	8.2	14.0	10.2	4.9	15.5
**2**	Andhra Pradesh	9.8	9.4	10.3	9.7	8.2	11.2	10.3	7.1	13.4	11.6	9.6	13.7
**3**	Arunachal Pradesh	8.3	7.1	9.5	6.9	5.2	8.7	8.6	4.7	12.5	8.7	4.6	12.9
**4**	Assam	5.5	3.2	7.9	4.1	1.5	6.7	5.7	3.4	8.0	5.9	2.9	8.9
**5**	Bihar	3.0	0.9	5.1	3.0	1.5	4.6	4.8	2.1	7.5	5.1	2.7	7.6
**6**	Chandigarh	10.0	20.0	0.0	11.0	22.0	0.0	17.3	34.6	0.0	12.9	25.6	0.2
**7**	Chhattisgarh	2.7	3.5	1.9	3.2	3.5	2.8	4.5	5.3	3.7	5.0	5.4	4.6
**8**	Dadra & Nagar Haveli	13.6	27.3	0.0	6.6	10.4	2.7	6.1	9.1	3.0	7.3	10.8	3.7
**9**	Daman and Diu	13.9	22.2	5.6	11.3	17.3	5.4	11.1	13.9	8.3	12.1	14.8	9.3
**10**	Delhi	5.6	0.0	11.1	8.6	0.4	16.8	3.5	0.0	7.0	11.8	0.6	23.0
**11**	Goa	7.1	0.0	14.3	11.3	17.9	4.7	14.4	13.6	15.2	12.0	13.1	11.0
**12**	Gujarat	6.2	7.5	4.8	6.2	7.3	5.1	7.0	8.1	5.9	7.9	8.2	7.6
**13**	Haryana	7.4	6.6	8.2	6.6	4.8	8.4	8.0	6.4	9.6	8.3	6.8	9.8
**14**	Himachal Pradesh	8.9	3.4	14.3	9.9	1.9	17.8	11.1	1.3	21.0	11.3	2.2	20.3
**15**	Jammu & Kashmir	8.5	3.7	13.3	9.2	3.9	14.4	10.3	6.1	14.6	10.6	5.5	15.8
**16**	Jharkhand	2.6	2.4	2.9	2.2	1.7	2.6	3.9	4.6	3.1	4.4	4.8	4.0
**17**	Karnataka	6.4	5.8	7.1	7.3	7.3	7.2	7.3	7.0	7.6	8.5	7.8	9.2
**18**	Kerala	17.0	12.8	21.3	11.0	10.5	11.4	13.7	11.5	15.8	13.2	10.6	15.8
**19**	Lakshadweep	9.1	18.2	0.0	13.0	25.9	0.0	15.2	28.3	2.2	13.8	22.6	5.0
**20**	Madhya Pradesh	3.6	4.4	2.7	3.4	3.0	3.9	5.8	7.6	3.9	5.6	5.8	5.4
**21**	Maharashtra	8.0	8.2	7.7	5.8	6.4	5.1	7.6	9.6	5.7	7.7	8.4	7.1
**22**	Manipur	7.4	5.2	9.7	8.4	6.1	10.7	10.6	10.4	10.8	10.8	9.3	12.2
**23**	Meghalaya	4.6	3.1	6.1	4.2	1.7	6.6	5.9	4.6	7.2	5.6	3.0	8.1
**24**	Mizoram	8.5	11.2	5.9	6.7	7.5	5.9	8.5	12.3	4.6	7.9	9.8	6.0
**25**	Nagaland	6.4	7.0	5.9	4.7	3.8	5.5	6.7	6.6	6.7	6.9	6.0	7.9
**26**	Odessa	5.7	3.1	8.2	4.0	2.6	5.4	6.6	5.1	8.0	7.0	4.8	9.1
**27**	Pondicherry	18.0	34.4	1.6	12.5	18.9	6.0	14.5	23.8	5.3	14.9	23.1	6.7
**28**	Punjab	15.5	4.6	26.4	9.8	7.1	12.5	12.1	9.4	14.8	11.6	8.7	14.5
**29**	Rajasthan	3.9	1.9	5.8	3.9	2.9	4.8	5.7	5.6	5.8	5.8	5.1	6.6
**30**	Sikkim	12.6	11.3	13.9	10.7	8.6	12.9	11.5	7.3	15.6	11.2	6.4	16.0
**31**	Tamil Nadu	12.2	11.9	12.6	10.3	10.5	10.1	11.1	11.4	10.8	11.9	12.6	11.3
**32**	Telangana	6.6	5.9	7.4	6.3	6.5	6.1	8.1	6.3	9.8	10.1	10.1	10.0
**33**	Tripura	6.8	4.6	9.1	5.7	3.2	8.2	5.1	6.0	4.1	7.3	6.1	8.5
**34**	Uttar Pradesh	5.4	3.2	7.6	5.6	4.6	6.5	6.8	5.6	7.9	6.8	5.6	8.0
**35**	Uttarakhand	8.3	5.4	11.2	7.2	7.5	6.9	7.4	6.3	8.5	7.7	7.0	8.5
**36**	West Bengal	5.0	3.7	6.3	5.4	4.0	6.7	6.9	8.0	5.9	8.0	6.9	9.1

Note: The prevalence presented in percentages.

**Table 3 ijerph-16-03987-t003:** The prevalence of obesity among adults according to age group, sex, and type of residence for each state and union territory.

State/Union Territory (Alphabetical Order)		18–34 Years						35–49 (54 Men) Years					
		Men			Women			Men			Women		
		Total	Rural	Urban	Total	Rural	Urban	Total	Rural	Urban	Total	Rural	Urban
**1**	Andaman & Nicobar Islands	0.0	0.0	0.0	2.1	0.7	3.6	2.7	2.2	3.3	3.2	1.4	5.1
**2**	Andhra Pradesh	3.8	5.1	2.6	2.2	2.5	1.9	3.8	4.8	2.9	4.1	4.3	3.8
**3**	Arunachal Pradesh	0.6	0.8	0.4	0.8	0.8	0.9	1.4	1.1	1.8	1.2	0.9	1.5
**4**	Assam	0.6	0.5	0.7	0.5	0.2	0.7	0.7	0.6	0.9	0.8	0.6	1.1
**5**	Bihar	0.5	0.1	0.9	0.6	0.2	1.1	0.9	0.6	1.1	1.0	0.7	1.3
**6**	Chandigarh	5.0	10.0	0.0	2.8	5.5	0.0	3.8	7.7	0.0	5.8	11.2	0.4
**7**	Chhattisgarh	0.2	0.0	0.3	0.5	0.6	0.4	0.8	1.2	0.3	1.1	1.4	0.7
**8**	Dadra & Nagar Haveli	4.5	9.1	0.0	0.8	1.6	0.0	0.8	1.5	0.0	2.4	4.5	0.4
**9**	Daman and Diu	0.0	0.0	0.0	0.6	1.2	0.0	2.5	3.0	2.1	4.4	5.4	3.3
**10**	Delhi	0.8	0.0	1.6	2.7	0.0	5.5	3.5	0.0	7.0	3.9	0.1	7.7
**11**	Goa	0.0	0.0	0.0	2.4	3.8	0.9	3.0	3.4	2.6	3.5	4.4	2.7
**12**	Gujarat	1.3	2.1	0.5	1.7	2.3	1.2	2.5	3.2	1.9	2.6	3.4	1.9
**13**	Haryana	2.1	2.5	1.6	1.0	0.8	1.3	1.9	2.0	1.8	2.0	2.0	2.1
**14**	Himachal Pradesh	2.0	0.0	4.0	1.9	0.6	3.2	2.7	0.6	4.8	2.6	0.6	4.7
**15**	Jammu & Kashmir	3.2	1.0	5.3	1.7	0.8	2.7	3.0	2.3	3.8	2.5	1.5	3.6
**16**	Jharkhand	0.3	0.3	0.3	0.4	0.4	0.5	0.5	0.8	0.2	1.0	1.4	0.6
**17**	Karnataka	1.4	1.7	1.0	1.4	2.0	0.8	1.8	2.0	1.7	2.3	2.7	1.9
**18**	Kerala	3.2	4.4	2.0	2.0	1.2	2.7	2.4	2.3	2.5	2.5	2.1	3.0
**19**	Lakshadweep	0.0	0.0	0.0	1.9	3.7	0.0	6.0	11.3	0.7	7.0	12.0	1.9
**20**	Madhya Pradesh	0.8	1.5	0.0	0.6	0.5	0.7	1.6	2.2	1.0	1.4	1.7	1.0
**21**	Maharashtra	2.2	3.9	0.6	1.0	1.1	0.8	2.2	3.0	1.4	2.0	2.5	1.4
**22**	Manipur	0.6	0.0	1.3	0.9	1.1	0.7	1.8	2.1	1.4	1.8	1.8	1.8
**23**	Meghalaya	0.6	0.4	0.7	0.5	0.2	0.8	0.6	0.7	0.4	0.7	0.6	0.8
**24**	Mizoram	0.6	0.6	0.6	0.8	1.0	0.6	1.2	1.7	0.7	1.1	1.4	0.7
**25**	Nagaland	0.7	1.3	0.0	0.6	0.6	0.7	1.1	1.7	0.6	0.8	0.8	0.8
**26**	Odessa	1.0	0.7	1.4	0.7	0.5	1.0	1.4	1.6	1.3	1.5	1.5	1.5
**27**	Pondicherry	4.9	8.2	1.6	2.9	4.3	1.4	4.3	7.8	0.8	4.3	6.6	2.1
**28**	Punjab	4.6	2.3	6.9	2.5	1.8	3.3	4.1	3.0	5.1	3.9	3.1	4.7
**29**	Rajasthan	0.6	0.6	0.7	0.7	0.7	0.7	1.4	1.8	1.0	1.4	1.7	1.2
**30**	Sikkim	1.0	0.7	1.3	1.4	1.3	1.6	2.1	2.0	2.2	2.2	1.9	2.5
**31**	Tamil Nadu	3.7	4.4	3.0	2.5	2.9	2.2	3.1	4.1	2.1	3.3	3.9	2.6
**32**	Telangana	1.5	0.0	2.9	1.7	2.0	1.4	2.8	3.7	2.0	3.0	4.2	1.9
**33**	Tripura	0.6	1.1	0.0	0.3	0.1	0.5	1.0	1.0	1.0	1.1	1.3	0.9
**34**	Uttar Pradesh	1.2	1.0	1.3	1.0	1.0	1.0	1.6	1.7	1.6	1.7	1.9	1.5
**35**	Uttarakhand	0.9	0.9	0.9	1.5	2.0	1.0	1.7	1.6	1.9	1.9	2.0	1.8
**36**	West Bengal	1.5	1.1	1.9	0.9	0.7	1.0	1.2	1.6	0.7	1.3	1.5	1.0

Note: The prevalence presented in percentages.

**Table 4 ijerph-16-03987-t004:** The prevalence of hypertension among adults according to age group, sex, and type of residence for each state and union territory.

State/Union Territory (Alphabetical Order)		18–34 Years						35–49 (54 Men) Years					
		Men			Women			Men			Women		
		Total	Rural	Urban	Total	Rural	Urban	Total	Rural	Urban	Total	Rural	Urban
**1**	Andaman & Nicobar Islands	1.0	0	2	0.71	0.4	1.1	3.02	2.5	3.56	2.65	0.8	4.5
**2**	Andhra Pradesh	0.4	0	0.9	1.01	0.5	1.5	3.09	2.3	3.92	2.71	1.9	3.5
**3**	Arunachal Pradesh	3.2	3.5	3	2.81	1.4	4.2	7.17	3.1	11.3	5.16	2.1	8.3
**4**	Assam	2.7	0.3	5.2	3.69	0.3	7.1	5.89	2.3	9.48	5.41	1.4	9.5
**5**	Bihar	1.0	0.3	1.7	0.89	0.2	1.6	1.94	0.4	3.43	1.67	0.5	2.9
**6**	Chandigarh	2.2	4.2	0.2	5	10	0	0	0	0	1.28	2.6	0
**7**	Chhattisgarh	2.1	1.3	2.8	1.4	0.5	2.3	3.29	2.2	4.41	2.93	1.7	4.2
**8**	Dadra & Nagar Haveli	0.0	0	0	1.1	2.2	0	3.79	4.6	3.03	2.24	2.4	2.1
**9**	Daman and Diu	0.0	0	0	1.2	0.6	1.8	2.67	3.9	1.48	2.07	2.5	1.6
**10**	Delhi	0.0	0	0	0.74	1.5	0	1.37	0.1	2.59	1.63	0	3.3
**11**	Goa	0.0	0	0	0	0	0	2.36	2.4	2.36	2.46	2.9	2
**12**	Gujarat	1.3	1.1	1.6	1.01	0.6	1.4	2.8	2.2	3.37	2.95	2.1	3.8
**13**	Haryana	2.5	2.5	2.5	1.33	1.1	1.5	2.67	2.6	2.78	2.65	2.1	3.2
**14**	Himachal Pradesh	0.3	0	0.6	1.13	0.4	1.9	3.35	0.3	6.39	3.02	0.4	5.6
**15**	Jammu & Kashmir	2.2	0.3	4	2.44	0.8	4.1	3.3	2	4.62	3.5	1.4	5.6
**16**	Jharkhand	1.2	0	2.4	1.15	0.3	2	2.91	1.2	4.62	2.35	1.2	3.5
**17**	Karnataka	1.0	0.3	1.7	1.51	1	2	2.97	2.2	3.72	2.85	2.3	3.4
**18**	Kerala	0.5	1.1	0	0.61	0.2	1	2.05	1.3	2.82	1.8	1.1	2.5
**19**	Lakshadweep	4.5	9.1	0	0	0	0	3.08	5.9	0.25	4.24	6.2	2.2
**20**	Madhya Pradesh	1.0	0.6	1.4	1.19	0.5	1.9	2.44	2.1	2.81	2.32	1.5	3.1
**21**	Maharashtra	1.4	0.6	2.2	0.74	0.4	1.1	2.63	2.3	2.98	2.16	1.8	2.6
**22**	Manipur	1.9	0	3.9	0.33	0.7	0	3.22	2.9	3.54	3.2	2.3	4.1
**23**	Meghalaya	1.1	0.6	1.7	1.9	0.3	3.5	2.57	1.2	3.98	2.91	1.1	4.7
**24**	Mizoram	1.5	1.8	1.2	1.81	1.2	2.4	2.66	3.1	2.22	2.1	2	2.2
**25**	Nagaland	2.7	3.2	2.1	2.12	1.2	3	6.83	6.9	6.79	5.4	3.9	6.9
**26**	Odessa	1.3	0.5	2.1	1.12	0.2	2	3.02	1.6	4.4	2.7	1.2	4.2
**27**	Pondicherry	1.6	3.3	0	1.58	2.6	0.6	2.57	4.2	0.95	2.74	4.4	1.1
**28**	Punjab	2.3	1.2	3.5	2.26	1.4	3.1	3.91	3	4.83	4.51	3.3	5.7
**29**	Rajasthan	1.0	0.1	1.9	0.8	0.4	1.2	2.11	1.5	2.75	1.98	1.3	2.7
**30**	Sikkim	2.7	3.3	2	3.31	1.8	4.9	5.24	3.3	7.15	5.23	2.7	7.8
**31**	Tamil Nadu	1.0	1.2	0.9	0.8	1.1	0.5	2.79	2.7	2.87	2.59	2.6	2.6
**32**	Telangana	1.8	0.7	2.9	0.83	0.7	0.9	3.11	2	4.22	2.61	2.2	3
**33**	Tripura	0.0	0	0	2.55	0.3	4.8	2.93	2.3	3.51	3.45	2.4	4.6
**34**	Uttar Pradesh	1.5	0.3	2.8	1.14	0.5	1.8	2.6	1.4	3.84	2.15	1.3	3
**35**	Uttarakhand	0.8	0.4	1.1	1.23	0.9	1.5	2.68	1.5	3.85	2.63	2	3.3
**36**	West Bengal	2.4	1.5	3.3	1.71	0.7	2.8	4.36	3.4	5.35	3.47	2.2	4.8

Note: The prevalence presented in percentages.

**Table 5 ijerph-16-03987-t005:** The prevalence of diabetes among adults according to age group, sex, and type of residence for each state and union territory.

State/Union Territory (Alphabetical Order)		18–34 Years						35–49 (54 Men) Years					
		Men			Women			Men			Women		
		Total	Rural	Urban	Total	Rural	Urban	Total	Rural	Urban	Total	Rural	Urban
**1**	Andaman & Nicobar Islands	3.1	0.0	6.1	3.2	1.1	5.3	5.2	4.4	6.0	3.8	1.8	5.8
**2**	Andhra Pradesh	8.1	6.0	10.3	5.0	4.3	5.7	6.8	4.8	8.8	7.0	6.3	7.7
**3**	Arunachal Pradesh	3.9	4.1	3.7	2.7	1.5	3.8	4.0	2.4	5.6	3.9	1.7	6.2
**4**	Assam	3.7	2.5	4.9	2.4	0.7	4.2	4.2	2.3	6.2	4.0	1.8	6.3
**5**	Bihar	2.1	0.3	3.9	1.8	0.5	3.1	2.7	0.6	4.8	2.6	1.0	4.2
**6**	Chandigarh	10.0	20.0	0.0	5.5	11.0	0.0	7.7	15.4	0.0	7.0	13.9	0.0
**7**	Chhattisgarh	2.7	1.3	4.1	2.1	1.1	3.1	3.2	2.5	3.9	3.4	2.2	4.5
**8**	Dadra & Nagar Haveli	0.0	0.0	0.0	2.5	3.3	1.6	1.5	1.5	1.5	3.3	3.4	3.2
**9**	Daman and Diu	2.8	5.6	0.0	4.2	5.4	3.0	5.9	7.4	4.4	6.7	9.0	4.5
**10**	Delhi	10.3	0.0	20.6	8.6	0.2	17.1	10.7	0.2	21.3	10.6	0.2	21.0
**11**	Goa	0.0	0.0	0.0	3.8	6.6	0.9	6.2	3.7	8.6	4.5	4.4	4.6
**12**	Gujarat	3.2	1.6	4.8	2.8	2.6	3.0	4.6	4.8	4.5	3.9	3.5	4.4
**13**	Haryana	2.9	0.8	4.9	2.4	2.1	2.8	2.6	2.0	3.2	3.1	2.4	3.7
**14**	Himachal Pradesh	2.3	0.6	4.0	2.7	0.8	4.7	3.5	0.6	6.5	3.7	0.9	6.6
**15**	Jammu & Kashmir	2.7	0.7	4.7	2.2	0.8	3.6	3.4	2.2	4.6	3.1	1.5	4.7
**16**	Jharkhand	1.8	1.0	2.6	2.4	1.3	3.5	3.1	2.0	4.2	3.3	2.5	4.2
**17**	Karnataka	3.1	3.4	2.7	2.1	2.0	2.3	4.6	3..45	4.6	4.0	3.4	4.6
**18**	Kerala	4.3	3.2	5.3	5.0	3.9	6.1	6.0	4.8	7.2	5.5	4.2	6.7
**19**	Lakshadweep	10.1	20.3	0.0	3.7	7.4	0.0	5.2	9.6	0.7	7.3	13.0	1.6
**20**	Madhya Pradesh	2.5	2.3	2.7	1.9	0.9	2.9	3.3	3.1	3.6	3.0	2.4	3.7
**21**	Maharashtra	3.0	2.8	3.3	2.0	1.9	2.1	3.7	3.7	3.7	3.5	3.5	3.5
**22**	Manipur	4.2	1.9	6.5	2.3	1.8	2.9	4.1	4.2	3.9	4.0	3.4	4.7
**23**	Meghalaya	2.7	1.7	3.7	2.5	0.7	4.2	4.1	2.0	6.2	3.3	1.8	4.8
**24**	Mizoram	3.5	4.7	2.4	3.1	2.3	3.8	5.6	7.4	3.9	4.6	5.0	4.1
**25**	Nagaland	5.4	5.4	5.4	3.4	2.3	4.4	5.6	4.7	6.5	5.3	3.5	7.1
**26**	Odessa	3.3	0.5	6.2	2.5	0.7	4.3	4.5	2.8	6.3	4.4	2.3	6.5
**27**	Pondicherry	1.6	3.3	0.0	2.4	2.9	2.0	4.5	6.9	2.2	4.7	7.0	2.5
**28**	Punjab	2.3	1.2	3.5	2.2	2.2	2.2	4.0	3.3	4.7	3.7	2.8	4.7
**29**	Rajasthan	2.5	0.3	4.7	1.5	0.6	2.4	2.4	2.0	2.8	2.3	1.6	3.1
**30**	Sikkim	2.3	2.7	2.0	2.6	2.0	3.3	4.0	3.2	4.8	3.8	2.1	5.5
**31**	Tamil Nadu	2.7	2.3	3.1	2.0	2.1	1.8	4.4	4.4	4.3	4.2	4.4	4.0
**32**	Telangana	6.3	2.9	9.6	3.2	2.7	3.7	5.9	5.1	6.7	6.4	6.8	6.0
**33**	Tripura	4.5	5.7	3.4	3.1	1.7	4.6	5.2	5.3	5.1	5.5	4.0	7.0
**34**	Uttar Pradesh	2.6	1.0	4.3	2.1	1.2	3.1	3.3	2.1	4.5	3.1	2.0	4.1
**35**	Uttarakhand	1.6	0.6	2.6	3.0	2.5	3.4	4.0	2.4	5.6	4.0	2.9	5.2
**36**	West Bengal	4.8	3.3	6.3	3.1	1.6	4.7	5.4	6.3	4.5	5.2	4.0	6.4

Note: The prevalence presented in percentages.

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
