# Peer review of "The Prevalence of Overweight, Obesity, Hypertension, and Diabetes in India: Analysis of the 2015–2016 National Family Health Survey"

_ijerph, 2019, doi:10.3390/ijerph16203987_

Round 1

Reviewer 1 Report

This is an important study which will be of high interest to many readers. Unfortunately the quality of the language and the presentation of the methods and results make it very difficult for me to assess to suitability of the methods used. I therefore highly recommend that you employ a copy editor to elevate the quality of the presentation. 

Author Response

Point 1. This is an important study which will be of high interest to many readers.

Response: Thank you for the reviewer for his constructive feedback.

Point 2. Unfortunately, the quality of the language and the presentation of the methods and results make it very difficult for me to assess the suitability of the methods used. I therefore highly recommend that you employ a copy editor to elevate the quality of the presentation.

Response: As suggested, we employed a copy editor to improve the quality of the presentation in this paper.

Reviewer 2 Report

Comments to the Authors

The topic of the  prevalence of overweight/obesity in various countries is relatively common. Therefore, the authors of the manuscript additionally provided information on the frequency of hypertension and diabetes in the population. The manuscript is good, but it still requires corrections:

The chapter Materials and Methods is too long and too boring. The authors of the manuscript described in detail the procedure for obtaining patients, but there is no description of research techniques, for example how the body weight and height were measured, and how the level of blood glucose was determinate. In the abovementioned chapter the authors wrote that to a group of overweight people were included those whose BMI was higher than 25 units. If in the study group were individuals under 18 years old, the use of such a cut-off point is incorrect. Therefore, persons under 18 years of age should be excluded from the study group. The collected material is large. The authors of the manuscript could separate two groups – overweight and obese individuals, and not combine them. The age range is extensive: 15 - 54 for men and 15 - 49 for women. In my opinion, two or three age groups should be distinguished here, within which statistical analysis would made. And as I mentioned above, people under 18 would be excluded from this analysis. The numbers of women and men participating in the study are given at the end of the chapter Materials and Methods. Therefore, in my opinion, it would be better to give results only in percentages in the chapter Results. Figure 1 is illegible. It should be broken into three independent maps. In tables 1 and 2, results should be presented as a percentage. Providing the results in the form of the number of individuals per 100,000 subjects, makes it difficult to analyze these results. The discussion is cursory. There is no analysis of the relationship between the prevalence of overweight and obesity and the place of residence, gender and there is no attempt to explain the reasons for the differentiation of these frequencies in the analyzed groups. The manuscript contains stylistic, letter and grammatical errors and in my opinion this text requires the correction of the native speaker.

Best regards!

Author Response

Point 1. The topic of the prevalence of overweight/obesity in various countries is relatively common. Therefore, the authors of the manuscript additionally provided information on the frequency of hypertension and diabetes in the population. The manuscript is good, but it still requires corrections:

Response: Thank you for the reviewer for his constructive feedback and suggestion.

Point 2. The chapter Materials and Methods is too long and too boring.

Response: This chapter is shortened by paraphrasing the sentences and focusing on the research question/hypothesis.

Point 3. The authors of the manuscript described in detail the procedure for obtaining patients, but there is no description of research techniques, for example how the body weight and height were measured, and how the level of blood glucose was determinate.

Response: Thank you for the reviewer for this important comment. The detailed descriptions of the techniques used to assess the selected biomarkers in this study are summarized in Table 1.

Point 4. In the chapter mentioned above, the authors wrote that to a group of overweight people were included those whose BMI was higher than 25 units. If in the study group were individuals under 18 years old, the use of such a cut-off point is incorrect. Therefore, persons under 18 years of age should be excluded from the study group.

Response: Thank you for the reviewer for this important comment. As suggested, we excluded the persons under 18 years of age from the study.

Point 5. The collected material is large. The authors of the manuscript could separate two groups – overweight and obese individuals, and not combine them.

Response: Thank you for the reviewer for this important suggestion. As suggested, we separated the two conditions as overweight and obese for individuals.

Point 6. The age range is extensive: 15 - 54 for men and 15 - 49 for women. In my opinion, two or three age groups should be distinguished here, within which statistical analysis would be made, and as I mentioned above, people under 18 would be excluded from this analysis.

Response: Thank you for the reviewer for this important suggestion. As suggested, we classified age into two groups: 18-34 years and 35-49 (54 for men) years after excluding the people under 18 years of age.

Point 7. The numbers of women and men participating in the study are given at the end of the chapter Materials and Methods. Therefore, in my opinion, it would be better to give results only in percentages in the chapter Results.

Response: As suggested, it has been modified accordingly.

Point 8. Figure 1 is illegible. It should be broken into three independent maps.

Response: As suggested, it has been modified accordingly and broken into four independent maps for each included biomarkers, such as for overweight, obesity, hypertension, and diabetes.

Point 9. In tables 1 and 2, results should be presented as a percentage. Providing the results in the form of the number of individuals per 100,000 subjects makes it difficult to analyze these results.

Response: We agree with the reviewer. As suggested, the results are presented as percentages for each biomarker in individual tables, such as table #3, #4, #5, and #6 for overweight, obesity, hypertension, and diabetes, respectively.

Point 10. The discussion is cursory. There is no analysis of the relationship between the prevalence of overweight and obesity, and the place of residence, gender and there is no attempt to explain the reasons for the differentiation of these frequencies in the analyzed groups.

Response: Thank you for the reviewer for this important comment. The discussion section has been modified accordingly.

Point 11. The manuscript contains a stylistic, letter, and grammatical errors, and in my opinion, this text requires the correction of the native speaker.

Response: As suggested, the text has been edited by the native speaker.

Reviewer 3 Report

The National Health Surveys are important to know the current health status of a certain population, which will allow us to establish public health strategies for prevention, management and treatment.

This is an interesting and essential work. This study provides information about the prevalence of overweight/obesity, hypertension and diabetes in the states and the union territories of Indian. These data were obtained from the fourth National Family Health Survey from January 2015 to December 2016. The results showed that 1 in 10, 76 and 19 individuals in India are overweight/obese, hypertension and diabetes. These results are important for future healthcare system and the policy of government.

-Specific

As this is the first study that reports the prevalence of these three conditions/diseases more information should be included.

In the introduction section, I think the authors should organize a bit better the ideas in this section of the paper. When the authors describe the prevalence data observed in other studies, some data are repetitive, I consider that the sentences should be shorter and more precise, and mention what this paper differs from the other studies already carried out. For example, regarding the prevalence of overweight/obesity, Shammi Luhar et al, BMJ Open, 2018., also reported prevalence of overweight/obesity, with data from the same National Family Health Survey in 26, 29 and 36 Indian states or union territories, in 1998/99, 2005/2006 and 2015/2016, respectively…?

The method is clear to me, and it is accordance to other National Family Health Survey reports from other countries.

Results

For future papers, I would recommend presenting the prevalence results in percentages, in this way, it would be easier to compare the results with studies or health surveys from other populations.

I would recommend to present the sociodemographic characteristics in a table, as a reader, I find it a bit difficult and confusing the way authors present these data.

As a suggestion, could we make some general analysis regarding the prevalence of overweight/obesity, hypertension and diabetes by socioeconomic level or by age groups? I think it would give more information about the current health status of the population in India, identifying groups currently most at risk of being overweight/obese, hypertensive or diabetic.

Discussion

As the introduction section, authors should organize the ideas in this section of the paper.

It is not the objective of the study, but I think a review could be made about the risk factors of the conditions reported in the study, in India. For example, Overweight/Obesity in Andhra Pradesh may be attributed to the rice-eating habits… That´s all, what about the sedentary lifestyle or high energy intake?

Generals

I think that the contribution of the authors to the scientific knowledge is that the data are presented by each of the Indian states?

I am not a native english speaker, but I consider that the english of the paper should be revised.

Author Response

Point 1. The National Health Surveys are important to know the current health status of a certain population, which will allow us to establish public health strategies for prevention, management, and treatment. This is an interesting and essential work. This study provides information about the prevalence of overweight/obesity, hypertension, and diabetes in the states and the union territories of Indian. These data were obtained from the fourth National Family Health Survey from January 2015 to December 2016. The results showed that 1 in 10, 76, and 19 individuals in India are overweight/obese, hypertension, and diabetes. These results are important for the future health care system and the policy of the government.

Response: Thank you for the reviewer for his constructive feedback.

Point 2. As this is the first study that reports the prevalence of these three conditions/diseases, more information should be included.

Response: As suggested, more information included about these three conditions/diseases and cited.

Point 3. In the introduction section, I think the authors should organize a bit better the ideas in this section of the paper.

Response: As suggested, the introduction has been modified accordingly.

Point 4. When the authors describe the prevalence data observed in other studies, some data are repetitive; I consider that the sentences should be shorter and more precise, and mention what this paper differs from the other studies already carried out. For example, regarding the prevalence of overweight/obesity, Shammi Luhar et al., BMJ Open, 2018., also reported the prevalence of overweight/obesity, with data from the same National Family Health Survey in 26, 29 and 36 Indian states or union territories, in 1998/99, 2005/2006 and 2015/2016, respectively…?

Response: Thank you for the reviewer for these important suggestions. As suggested, the sentences are shortened by removing the repetitive text or data and mentioned how this paper differs from the other studies already carried out, including the study by Shammi Luhar et al. (2018).

Point 5. The method is clear to me, and it is in accordance with other National Family Health Survey reports from other countries.

Response: Thank you for the reviewer for his supportive comment. However, the method has been shortened as per the comment from another reviewer by focusing on the research question/hypothesis.

Point 6. For future papers, I would recommend presenting the prevalence results in percentages; in this way, it would be easier to compare the results with studies or health surveys from other populations.

Response: Thank you for the reviewer for his supportive recommendation. However, the prevalence results are summarized in percentages as per the comment from another reviewer.

Point 7. I would recommend presenting the sociodemographic characteristics in a table; as a reader, I find it a bit difficult and confusing the way authors present these data.

Response: Thank you for the reviewer for this important recommendation. As suggested, the sociodemographic characteristics of the study sample are presented in Table #1.

Point 8. As a suggestion, could we make some general analysis regarding the prevalence of overweight/obesity, hypertension, and diabetes by socioeconomic level or by age groups? I think it would give more information about the current health status of the population in India, identifying groups currently most at risk of being overweight/obese, hypertensive or diabetic.

Response: Thank you for the reviewer for his important suggestion. We agree. As suggested, the analysis has been done by age group classifyingng into two groups: 18-34 and 35-49 (54 for men).

Point 9. As the introduction section, authors should organize the ideas in the discussion section of the paper.

Response: As suggested, the discussion section has been modified accordingly as the introduction section.

Point 10. It is not the objective of the study, but I think a review could be made about the risk factors of the conditions reported in the study, in India. For example, Overweight/Obesity in Andhra Pradesh may be attributed to the rice-eating habits… That´s all. What about the sedentary lifestyle or high energy intake?

Response: As suggested, the review made about the risk factors of each condition/disease and reported accordingly; even it is not the objective of the study as said.

Point 11. I think that the contribution of the authors to scientific knowledge is that the data are presented by each of the Indian states?

Response: Thank you for the reviewer for his constructive feedback about the scientific knowledge of the paper.

Point 12. I am not a native English speaker, but I consider that the English of the paper should be revised.

Response: As suggested, this paper has been edited by the native English speaker.

Round 2

Reviewer 2 Report

The manuscript was corrected in accordance with the recommendations of reviewers and I have no more comments about this article.

Author Response

Point 1: The manuscript was corrected in accordance with the recommendations of reviewers, and I have no more comments about this article.

Response: Thank you for the reviewer for his constructive feedback on our revisions.

Reviewer 3 Report

As a whole, authors had addressed the comments and suggestions made. They made important changes.

As a reader I still find it a bit confusing the way the results were presented.

Table 2 shows the sociodemographic characteristics, however, the authors added 3 clinical indicators that would not correspond (BMI, BP and Glucose), perhaps change the title to “general characteristics of the study sample” and add a subtitle to separate them “clinical indicators ”.

The authors present the results of the prevalence of clinical conditions by states of India in different ways.

Total prevalence (%) by age group in Figure 3.

Total prevalence per 100,000 thousand / hab figure 4, 5, 6 and 7.

Prevalence by age group, sex and areas (rural urban) in table 3, 4, 5, 6

I think it is the same information presented only in different ways. Suggestion, we could ignore figure 3 and include a column with “total prevalence” in tables 3, 4, 5, 6.

General, can we improve the design? The figures are not very clear.

General, to be considered by the authors, can we present the results with just one decimal, not two.

In general, indicates as a footer in the tables or figures how the prevalence is presented, in this case in (%)

Author Response

Point 1: As a whole, the authors had addressed the comments and suggestions made. They made important changes.

Response: Thank you for the reviewer for his constructive feedback on our revisions.

Point 2: As a reader, I still find it a bit confusing the way the results were presented.

Response: Thank you for the reviewer for his comment on the presentation of results.

Point 3: Table 2 shows the sociodemographic characteristics; however, the authors added 3 clinical indicators that would not correspond (BMI, BP, and Glucose), perhaps change the title to “general characteristics of the study sample” and add a subtitle to separate them “clinical indicators.”

Response: Thank you for the reviewer for his comments. As he suggested, it has been modified accordingly in the table #2.

Point 4: The authors present the results of the prevalence of clinical conditions by states of India in different ways. Total prevalence (%) by age group in Figure 3. Total prevalence per 100,000 thousand had figures 4, 5, 6, and 7. Prevalence by age group, sex and areas (rural-urban) in tables 3, 4, 5, 6. I think it is the same information presented only in different ways. Suggestion, we could ignore figure 3 and include a column with “total prevalence” in tables 3, 4, 5, 6.

Response: Thank you for the reviewer for his valuable comments. We agree with him that the same information was presented in different ways. However, we deleted the figure #3, as recommended by the reviewer. Also, a column with total prevalence was incorporated in the tables 3, 4, 5 and 6.

Point 5: General, can we improve the design? The figures are not very clear.

Response: As suggested by the reviewer, each figures’ design was improved.

Point 6: General, to be considered by the authors, can we present the results with just one decimal, not two.

Response: As suggested by the reviewer, the results were presented with just one decimal.

Point 7: In general, indicates as a footer in the tables or figures how the prevalence is presented, in this case in (%).

Response: As suggested by the reviewer, the footer has been noted accordingly in the tables’ #3, 4, 5 and 6, and figure #2.